# Recurring Cystitis: How Can We Do Our Best to Help Patients Help Themselves?

**DOI:** 10.3390/antibiotics11020269

**Published:** 2022-02-18

**Authors:** Sarah Ben Hadj Messaoud, Elisa Demonchy, Véronique Mondain

**Affiliations:** Infectious Diseases Department, Université Côte d’Azur, Centre Hospitalier Universitaire de Nice, 06000 Nice, France; sarah.benhadjm@gmail.com (S.B.H.M.); demonchy.e@chu-nice.fr (E.D.)

**Keywords:** recurrent cystitis, urinary tract infection, antibiotic prescription, non-antibiotic treatment

## Abstract

Recurrent cystitis (RC) has rarely been studied; its management varies and research on a holistic approach of these patients is scarce. We attempted to characterize patients suffering from RC and investigated their current care pathways, aiming to offer customized and autonomous management. In this paper, we present a descriptive, single-center, cross-sectional study of women presenting with RC at an infectious disease (ID) clinic. A questionnaire was developed and was completed by ID physicians during patient visits. From October 2016 to January 2019, 202 women were included (mean age 59 years). Sexual intercourse, stress and diarrhoea/digestive symptoms were reported as trigger factors by 35%, 34% and 19% of patients, respectively. A majority (54%) were at risk for complications and were those more exposed to inappropriate antibiotic prescriptions. In total, 56% of women suffered from more than 10 episodes/year and 48% suffered from relapses, mainly due to *E. coli*. Genitourinary syndrome of menopause (GSM) was a frequent complaint (74.5% of women). Fluoroquinolones and 3rd generation cephalosporins were prescribed in 38% and 30% of women, respectively. Most women wished for non-antimicrobial approaches and autonomy. Patients require comprehensive, tailored care in order to benefit from a broader range of treatment options in compliance with recommendations.

## 1. Introduction

In France, in 2016, urinary tract infections (UTI) were estimated to account for a third of all antibiotic prescriptions, most of which were provided by general practitioners (GP) [1]. That year, the societal cost of UTI in France was estimated at €58 million, of which 25% were associated with negative cultures [2].

In the USA, the prevalence rate for these infections in women is over 50%, with approximately 3% developing recurring cystitis (RC), i.e., at least four episodes a year [3].

To our knowledge, research on RC in France has, until now, been limited. A study conducted in 2006 concluded that 2.3% to 10% of women suffered from this condition, but little is known of patients at risk for complications, and no specific recommendations have been published for these patients [4].

Management of RC is highly variable both within and between countries [5]. Case definition and diagnosis are not clear-cut, as clinical and biological factors and/or overall risk factors are not considered. This results in a monolithic approach involving iterative antibiotic treatment courses, leading to emergence of antimicrobial-resistant bacteria and significant side-effects [6]. Women in many countries appear unhappy with the way RC is managed [7].

The French language infectious disease society (Société de Pathologie Infectieuse de Langue Française, SPILF) updated its recommendations in 2015 and 2017, excluding the use of third-generation cephalosporins (3GC), amoxicillin-clavulanate combinations, and fluoroquinolones for the management of uncomplicated acute cystitis and those at risk of complications [8,9]. However, compliance with recommendations is inadequate; antibiotic prescription remains inappropriate, with persistent use of critically important compounds [10].

Most scientific research on RC has focused on pathological processes, antimicrobial treatment and its alternatives, antibacterial resistance and inappropriate antibiotic use. However, patients’ behavioural and psychological determinants have not been extensively explored. Although cystitis is considered a mild condition that can cure spontaneously in 40% of cases when uncomplicated [11], it exerts a major impact on women’s quality of life [12]. Little is known about the use of health care services by these patients, while consequences for society are significant [2].

To investigate and improve practices, a working group was established within the PACA-Est Infectious Disease network.

The present study, thus, aimed to identify these patients’ profiles and to describe their care pathways as a preliminary step, prior to suggesting a customized management strategy to be subsequently assessed.

## 2. Method

This was a single-centre cross-sectional study conducted on patients who presented with recurrent cystitis at the infectious diseases department outpatient clinic of Nice University Hospital between 19 October 2016 and 15 January 2019.

### 2.1. Study Population

Female patients with RC who presented at the infectious disease outpatient department of Nice University Hospital were recruited.

### 2.2. Data Collection

A questionnaire was developed, (See Appendix A) based on the SPILF recommendations and the propositions put forward by the working group and was validated by the PACA-Est Infectious Diseases Network, comprising GPs, infectious disease specialists (ID), urologists, gynaecologists, pharmacists and biologists.

The questionnaire was given to the patient as she arrived at the outpatient clinic, allowing her the time to reflect on the questions relating to RC. It was then completed with the 2 ID specialists who were consulted by these women. The resulting database included study population characteristics: age group, socio-professional category, perceived psychological profile, patient history (age of onset of cystitis, circumstances of occurrence, trigger factors, complications), clinical signs, risk factors (family history, risk behaviour, hormonal status), risk factors for complications, and bacterial ecology within the cohort.

The questionnaire also explored ambulatory care pathways: management, investigation procedures and treatment of RC.

Ethical approval was not required for this study, as it was an internal single-centre institutional research project in line with the general information provided to all patients on data utilisation. A standard institutional document explaining this use is systematically signed by patients visiting the institution. The study has been entered in the institutional register of data processing activity under number 2022033CHUN.

### 2.3. Statistical Analysis

Data were entered and analysed, and graphs constructed using MS Excel software. Descriptive statistical analysis was performed with variables shown as N and %.

### 2.4. Results

Two hundred and two patients were recruited between 19 October 2016 and 15 January 2019. The first 54 patients completed the questionnaire with the ID specialist during the consultation, as it was still not available as a hand-out. The remaining 148 questionnaires had been given to the patients just before the consultation, thus, allowing time for them to read and reflect upon the questions before discussing them with the ID specialist.

Mean patients’ age was 59 years (range: 17 to 90 years), 71% were over 50, 58% were retired and 79% were urban residents, 74.5% were in perimenopause or menopause. Symptomatic anxiety was described by 62% of patients, 7% of whom stated that they were depressed. Most women had at least one child, while 26% were nulliparous.

A majority (62.5%) of women described a history of cystitis of at least ten years’ duration, dating back to childhood for 12% of them. Frequency of cystitis had increased over the past three years for 55% of patients; 43% stated that this had been the case since menopause. Over half the patients (55%) had more than 10 episodes of cystitis every year. The main trigger factors included sexual intercourse (36%) and stress (35%), followed by diarrhoea (19%). Forty percent mentioned a history of acute pyelonephritis. Clinical signs reported by patients included urinary frequency (72%), burning sensation upon micturition (70%), suprapubic pain (65%), smelly urine (22%) urinary incontinence (18%), hematuria (17%) and cloudy urine (14%).

Risk factors for an episode of cystitis are described in Table 1, while those for complications are shown in Table 2.

A urine culture was prescribed at each episode for 65% of patients, while this was occasional for 32%. Follow-up urine cultures were performed in 20% of patients (Table 3).

Results of urine cultures provided by patients showed that *E. coli*, *Klebsiella* spp. and *Enterococcus* spp. accounted for 88%, 28% and 21% of cultured samples, respectively. Extended-spectrum beta-lactamase-producing Enterobacteriaceae were identified in 17% of isolates.

Antimicrobial agents prescribed during an acute episode are shown in Figure 1. Treatment was most often prescribed empirically and rarely readjusted according to urine culture results. Treatment was considered fully effective by 40.5% of patients, variably effective according to the episode by 53.0% and ineffective by 6.5%.

With regard to the recommendations put forward by the French Language Society of Infectious Diseases (SPILF), which states that 3rd generation cephalosporins and quinolones should be avoided in cystitis, antibiotic treatment was appropriate for 18.5% of patients, variably appropriate according to the episode for 55.0% and inappropriate for 26.5%: there were 69 fluoroquinolone prescriptions and 55 3rd generation cephalosporin prescriptions (Figure 1).

Of 90 documented prescriptions for antibiotic prophylaxis, 42% were appropriate: weekly fosfomycin-trometamol (FT) or weekly co-trimoxazole (TMP-SXT) (Figure 2). Treatment was considered fully effective by 15% of patients questioned, variably effective by 12% and ineffective by 57%.

Prior to seeking infectious diseases specialist advice, 79% of women had consulted a urologist and 74% a gynaecologist. Examination of the perineum was not systematic; patients were not always informed of the conclusion. Investigations undergone by patients are shown in Figure 3.

According to the SPILF recommendations, the following investigations were considered inappropriate for the patients considered: cystoscopy (*n* = 12), MRI (*n* = 6), urology CT scan (*n* = 8), cystography (*n* = 4).

Antibiotic treatment resulted in adverse events in 37% of patients, consisting in gastro-intestinal disturbances, frequent vaginal candidiasis, and cutaneous reactions. Four women presented with fluoroquinolone-related tendinopathy, one with toxidermia and one developed Lyell’s syndrome.

Most women (70%) resorted to non-antibiotic therapies, mainly plant-based compounds and particularly proanthocyanidine (cranberry). Other approaches included homeopathy, D mannose, acupuncture, hypnosis, naturopathy, Chinese medicine, essential oils and mesotherapy. However, their use was highly variable, both in duration and modality (acute episode or background treatment).

Most patients (53%) were referred to the infectious diseases department by their general practitioner, the remainder by urologists (24%), gynaecologists (8%), or other specialists (15%).

### 2.5. Patient Categories

Based on collected data, the following patient categories were identified (Figure 4)

Simple RC (29%), RC at risk for complications (54%), post-coïtal cystitis (32%), recurring cystitis due to different microbial agents (24%), relapsing cystitis due to the same microbial agent (48%), chronic RC (56%) arbitrarily defined as >10 cystitis episodes/year.

There was some overlap between these categories, as 60% of women at risk of complications had chronic RC and 58% had relapses. Antibiotic treatment was inappropriate in 85% of acute cases and 35% in prophylaxis.

For those women considered as having simple RC, 46.5% had chronic RC and 55% had relapses. Antibiotic treatment was inappropriate in 72% of acute cases and 22% in prophylaxis.

Many patients considered to be suffering from chronic RC had relapses. Bacterial documentation was lacking for 11% of women. For those reporting post-coital RC, the only contributing factor was sexual intercourse in 44% of cases.

### 2.6. Differential and Associated Diagnoses

Referral to the infectious diseases department led to the conclusion that 17% of women did not meet criteria for RC but presented with symptoms of urinary colonisation and/or sterile urine culture.

Among those with definite RC, 28% were found to have an associated diagnosis: painful bladder syndrome (PBS: 19%), suspected interstitial cystitis (IC: 8%) or confirmed by cystoscopy (3.5%), urethral syndrome (10.5%), vulvar/vaginal pain (7%). Almost half of those with PBS or IC complained of bowel disorders.

## 3. Discussion

The present study collected data from a variety of women with RC, regardless of age and risk factors for complications, including contributing factors, behavioural aspects and psychological impact. A major finding was the chronic state of anxiety revealed by many women and the importance of stress as a trigger factor. These are not considered in recent recommendations [8,9]. Few studies describe such features as contributing factors. However, Reese et al. had already reported, in 1977, that 30% of women with cystitis had a history of anxiety and/or depression antedating micturition symptoms [13]. Anxious, obsessive personalities with somatic expression of their psychological/emotional disorder may prevail among women with RC, interfering with complete voiding, and the authors suggest that a multifactorial approach, including awareness of psychiatric factors, will reduce treatment failure rate. It has been hypothesized that activation of the autonomous nervous system through stress could result in relaxation of the detrusor muscle and contraction of the smooth muscles of the urethral sphincter, leading to urinary retention and, hence, to RC. Research on the potential benefits of hypnosis is currently ongoing at the Nice University hospital infectious diseases department, an approach supported by the stress and neuroticism observed among patients. Such a broader approach is increasingly favoured, addressing both somatic and psychological aspects of the condition, while improving women’s well-being, reducing inappropriate antibiotic use, bacterial resistance and, potentially, costs [14,15]. Fan et al. explored the psychological profile of women with urinary dysfunction and found that 59% were depressed [16]. Other authors suggest that neuroticism, i.e., a tendency to experience negative feelings such as anxiety and guilt, might be more frequent in patients with RC and influence patients’ behaviour as a result of increased sensitivity to their surrounding environment in relation to a particularly reactive sympathetic nervous system [17]. The authors advise taking this trait into consideration to widen the approach to management, which should include both organic and psychological factors. Sexual violence was a delicate subject to bring up, but 2 of the 11 patients questioned admitted having been subjected to it, and, in view of its considerably distressing impact, screening should be extended [18,19]. Indeed, immune dysregulation has been shown among adults who experienced stressful situations during childhood [20].

Over half the patients stated they had at least ten episodes per year, showing the extent of the impact on their professional and personal life. Recent studies have shown that physical discomfort and psychological distress affected women’s social and professional activities and intimate life, with 17% of women renouncing sexual intercourse due to its link with cystitis [21,22,23]. Confronted with antimicrobial treatment failures and the perceived lack of concern on the part of health professionals, women felt disparaged, and their response was one of resignation [21].

In view of the above-mentioned multi-faceted features of RC, and the shortcomings of routine medical management, most women resorted to non-antibiotic approaches: plant extracts, probiotics, D-mannose and Chinese medicine [24,25]. Proanthocyanidin, a polyphenol extracted from cranberries, has been shown to prevent *E. coli* from adhering to the urethral epithelium, to exert an anti-inflammatory effect and to reduce urine pH [26,27]. Although the effectiveness of such approaches has yet to be clearly demonstrated, avoidance of antimicrobial resistance resulting from iterative antibiotic courses argues in their favour for many patients, providing that prescriptions are appropriate and innocuous [28].

Relapsing *E. coli* infection was frequent, suggesting permanent colonisation of the bladder by the pathogen in its intracellular and quiescent form, escaping immune response and antimicrobial activity [29,30]. Indeed, uropathogenic *E. coli* possess type 1 pili, allowing them to penetrate the bladder epithelium, a property that can be countered by D-mannose [31,32]. Host genetic characteristics also play a major role in the immune response to infection, namely Toll-like receptor variability [33].

Among our patient cohort, 17% were carriers of extended-spectrum beta-lactamase-producing *Enterobacteriaceae*, i.e., four-fold the average rate in France among women with UTI [34]. Multi-resistant bacteria currently account yearly for 125,000 infections and 5500 deaths in France [35]. Given the proportion of inappropriate prescriptions in our patient cohort, efforts towards improving management of RC appear essential.

Over 50% of patients were at risk for complications, among whom a quarter were over 75 years of age. In the absence of any specific recommendation for this patient category, these women were particularly exposed to inappropriate antimicrobial treatment.

A history of acute pyelonephritis was reported for 40% of the cohort, regardless of risk factors for complications. Eto et al. suggest that patients with RC have a higher risk of developing acute pyelonephritis, although the high proportion observed among our patients may be due to a selection bias, since they had been referred to an infectious disease specialist [29].

Some of the risk factors for RC mentioned by patients could be corrected: excessive intimate hygiene disrupting the protective lipid barrier, insufficient hydration, consumption of irritants, retained micturition or voiding in a non-sitting posture. Most complained of intestinal disorders, (diarrhoea was stated as a trigger factor more often than constipation), vaginal dryness and pelvic floor dysfunction. Post-coital RC was mentioned regardless of age. Most patients had undergone menopause. According to international studies, over 50% of women suffer from genitourinary syndrome of menopause (GSM), i.e., atrophic vaginitis and increased vaginal pH related to oestrogen deprivation, with genital, urological and sexual symptoms that patients are reluctant or embarrassed to discuss, when appropriate management would improve quality of life and prevent complications, including urinary urgency and incontinence [36,37].

Two thirds of women who had undergone sling surgery for urinary incontinence or prolapse reported complications, considering that the procedure had triggered the onset of RC. Concern over the innocuity of these procedures has led to current monitoring of subsequent complications [38].

Lastly, a number of women reported chronic pelvic pain, i.e., painful bladder syndrome, interstitial cystitis, urethral syndrome and/or chronic vulvodynia in association with RC, for which a specific aetiology is rarely identified, and which can benefit from a patient-centred approach, taking the psychological component into account. A smartphone application has been developed for these women and a similar one dedicated to RC might prove useful [39].

Although current recommendations state that infectious disease specialist advice should be sought for women, patients in the present study were often referred several years after their first episodes of RC. In the present study, the diagnosis was unfounded in 17% of women. Aside from typical clinical signs, the questionnaire investigated the presence of “cloudy” or “smelly” urine, neither of which necessarily point to a UTI, providing the opportunity to inform patients of true symptoms of infection.

Many women were prescribed a fosfomycin-trometamol combination. However, fluoroquinolones and 3GC were prescribed to 38% and 30% of patients during an acute episode, respectively. Urine dipstick tests were rarely performed in this cohort, and urinalysis was prescribed to most women, according to recommendations, and repeated following antimicrobial treatment for at least 20% of them, which is not recommended, as this could lead to prescription of a second-line antibiotic in the case of persistent bacteriuria, regardless of symptomatology. Indeed, urinary bacterial colonisation still leads to numerous unnecessary antibiotic prescriptions. Antimicrobial prophylaxis, which included fluoroquinolones (6% of patients) and 3GC (6% of patients), was not consistently appropriate and was judged unsatisfactory by a majority of women, with 37% reporting adverse events.

Among our patients, many had unnecessarily undergone a cystoscopy, which was not recommended in their case. Furthermore, urologists had not given patients the results of the procedure. The SPILF recommendations state that, for women with recurrent cystitis who are not at risk for complications, investigations should be limited to clinical pelvic and ultrasound examination. In the case of risk factors for complications, investigations should be decided according to the clinical situation.

Thus, there is a broad scope for improving the management of RC. Systematic screening and consideration of all these factors would assist the physician in determining a specific and coherent management strategy thanks to a decision-making algorithm that would include both somatic and psychological/behavioural characteristics while complying with good clinical practice recommendations. This should involve a dedicated medical consultation and identification of expert referees, while empowering and educating women to achieve self-care, based on their needs and expectations. Anticipatory prescriptions for urinalysis, provision of clear guidelines in case of cystitis, and education on self-treatment would contribute to achieve this. An information leaflet has, thus, been developed to provide patients with better knowledge of their condition and more autonomy to manage it.

This study has several limitations. A recall bias was likely, as some women had a protracted history of RC and could not always remember the antibiotic compounds or investigations they had been prescribed. Not all could provide informative documentation. Furthermore, women who presented at the infectious diseases outpatient department were often referred by urologists who were aware of issues related to RC, potentially also resulting in a selection bias.

The present study points out the current failings in the management of RC. It reveals excessive, unnecessary antibiotic prescriptions, including critically important compounds, resulting in over a three-fold increase in multi-drug-resistant *E. coli*, compared to the rate of 4.5% in community samples in France [40], a lack of treatment re-assessment based on urine culture results following an initial empirical prescription, and superfluous post-treatment urine cultures, increasing the financial burden of the condition.

The issue addressed in our manuscript does not appear restricted to France: a survey conducted in England among patients and GPs showed that (1) management of recurrent cystitis was often unsatisfactory for women due to their lack of communication with their GP, (2) antibiotic stewardship could be improved and (3) patient empowerment was desirable for self-management of their condition [7]. Similarly, a study conducted in five countries highlighted the impact of RC on women’s quality of life [12]. In Canada, recent efforts have resulted in better characterization of patients and in improved antibiotic stewardship with fewer and more appropriate prescriptions, although ciprofloxacin remains the second most prescribed antibiotic for RC [41]. Furthermore, among patients who visited the ID department, some came from various regions of France, but also from Spain or Italy, expressing similar complaints.

The state of stress and anxiety related to recurrent cystitis, whether as its cause or its consequence, the pain accompanying each episode and the resulting disruption of daily life are usually underestimated by health professionals as well as by society. Women resort to various alternative treatments in an attempt to manage their condition, while avoiding iterative antibiotic courses that they increasingly resent. While treatment of cystitis is relatively simple and straightforward, the aim should be to treat RC using an approach that is as respectful as possible of the microbial ecosystem through appropriate antibiotic stewardship while avoiding secondary infection, i.e., to solve the problem as best as possible with the least risk. At the same time, this approach must take into account that women currently have to wait 5 to 7 days from the time they take their urine sample to the lab, obtain the result, and see their GP to get a prescription, a complex and lengthy process during which their pain and discomfort significantly impair their quality of life. This could be shortened if they were given more autonomy, advice and means of self-care. Furthermore, in view of the range of patient categories, it is important to consider the highly multifactorial aspect of cystitis that should, thus, require a tailored and holistic approach, based on relevant recommendations, taking into account its psychological aspects as well as the microbial ecosystem.

Such an autonomous, ecological approach should be encouraged by providing appropriate advice and adequate, caring and empathic support on the part of their care-providers.

## Figures and Tables

**Figure 1 antibiotics-11-00269-f001:**
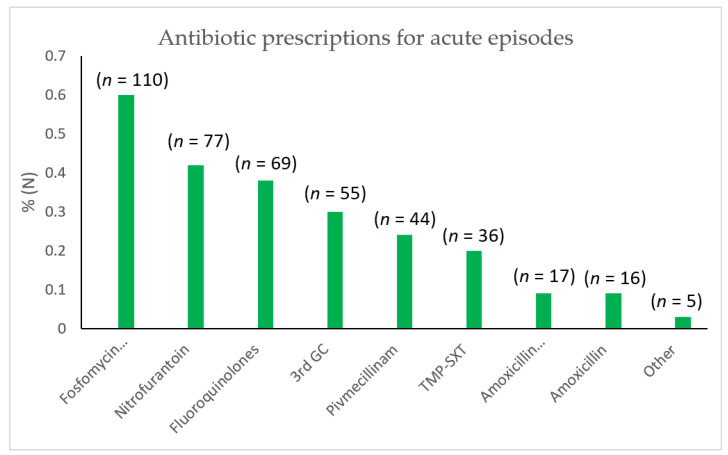
Distribution of prescribed antimicrobial agents (3rd GC: third generation cephalosporins).

**Figure 2 antibiotics-11-00269-f002:**
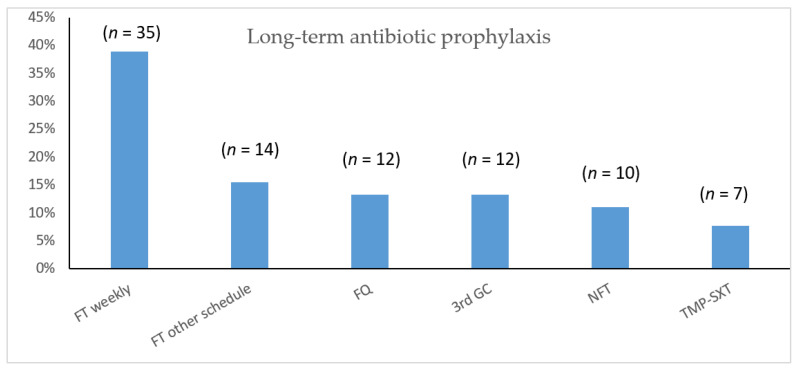
Long-term prophylactic antibiotic prescriptions, FT: Fosfomycin trometamol; FQ: fluoroquinolones; 3rd GC: 3rd generation cephalosporins; NFT: nitrofurantoine; TMP-SXT: trimethoprim-sulfamethoxazole.

**Figure 3 antibiotics-11-00269-f003:**
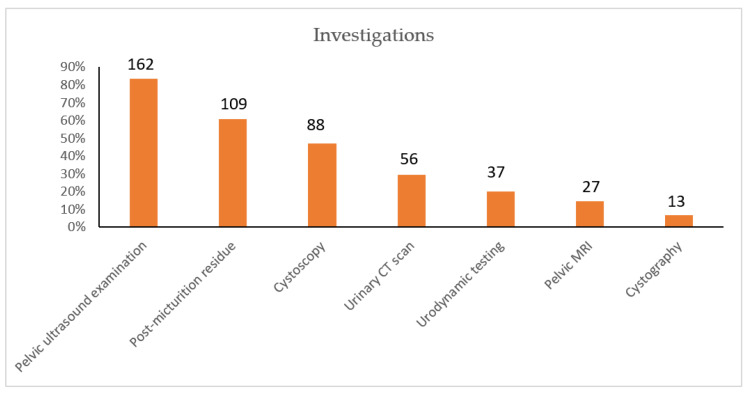
Investigations undergone by patients with recurring cystitis.

**Figure 4 antibiotics-11-00269-f004:**
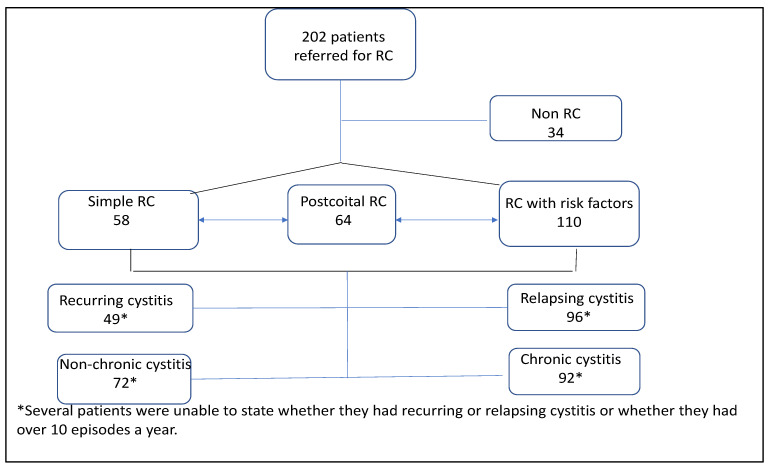
Patient distribution according to RC category.

**Table 1 antibiotics-11-00269-t001:** Factors identified as favouring cystitis when questioning patients.

Favouring Factors	N	%
Trigger Factors:		
Sexual intercourse	73	36%
Stress	71	35%
Diarrhoea	38	19%
Behavioural factors		
Anxiety	125	62%
Hydration < 1.5 L/day	65	32%
Bladder irritants:		
Excess intake of tea, coffee, alcohol	38	19%
Tobacco	25	13%
Withheld or non-seated micturition	55	28.5%
Excessive intimate hygiene ≥2x/day	78	40.5%
Inadequate drying	5	2.5%
Aggravating sports (cycling, horse-riding…)	16	9%
Obesity	23	12%
Non-behavioural factors		
Family history of cystitis	56	28%
Irritable bowel syndrome	105	52%
Constipation	50	25%
Diarrhoea	27	14%
Alternating diarrhoea/constipation	30	15%
Perceived vaginal dryness	121	60%
Non-perceived vaginal dryness	23	12%

**Table 2 antibiotics-11-00269-t002:** Risk factors for complications.

Risk Factors for Complications	N	%
Age > 75 years	49	24%
Organic abnormalities of the urinary tract	42	22%
Hymenal adhesions	13	7.5%
Urolithiasis	25	13%
Bladder diverticula	4	2%
Dilated urethral stenosis	20	10%
Non-operated pelvic floor dysfunction	18	10%
Sling surgery—Prolapse cure	49	24%
Among which ineffective and/or with complications (34/49)	34	69%
Other urological surgery	25	12%
Functional abnormalities of the urinary tract		
Residual urine > 100 mL	32	16%
Neurogenic bladder	7	3%
Hyperactive bladder	10	6%
Dystonic urethral sphincter	8	5%
Medical treatment favouring post-micturition bladder residue	48	24%
Among which those with proven bladder residue (8/48)	8	16%
Iatrogenic immune depression:		
methotrexate (8), steroids (9), immune suppressors (11), immune modulators (1), monoclonal antibodies (7)	27	13%
Non-iatrogenic immune depression:		
amyloidosis, hypogammaglobulinemia, HIV	3	1%
Pelvic radiotherapy	3	1%

**Table 3 antibiotics-11-00269-t003:** Frequency of urine culture and urinalysis.

Urine Culture and Urinalysis	N	%
Never	6	3%
Sometimes	65	32%
Always	131	65%
Post-treatment urine culture and urinalysis		
Never	161	80%
Sometimes	17	8%
Always	12	12%

## Data Availability

The data presented in this study are available on request from the corresponding author.

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
