# Peer review of "Recurring Cystitis: How Can We Do Our Best to Help Patients Help Themselves?"

_antibiotics, 2022, doi:10.3390/antibiotics11020269_

Round 1

Reviewer 1 Report

This manuscript is a survey completed about women who have recurrent cystitis at one clinic over the course of ~2 years. The goal was to characterize the patients and their management. The authors should be commended for their efforts in capturing this data so that we know more about this difficult-to-manage patient group.

My biggest concern about the manuscript is that its utility is likely limited to the authors’ country of France (and maybe just limited to their center too).

My next big concern is that the Methods are not very clear. In the abstract, the authors claim this is a cohort study, but it seems like the questionnaire was only completed one time for each patient, which is more indicative of a cross-sectional design. Additionally, were the questionnaires completed by asking at least some of the questions to the patients? Or solely by the knowledge of the physicians? This should be clarified.

My third concern is the first paragraph in Results coupled with the ethical approval statement. If at least some of the information on the questionnaire needed to be completed by talking with the patient, then why were 54 responses included when they only used the medical files? And why were the questionnaires not completed with the patients? This concerns me that these patients might not have wanted their data collected and analyzed, even in anonymous form. This is especially concerning with the statement that ethical approval was not required. Was ethical approval sought and deemed “exempt” or not required? Or was it not sought at all? Assuming ethical approval was sought and deemed not required, then one way to modify this is to re-run the analysis with only the 148 questionnaires that were completed with the patients. I’m further confused by the statement in Discussion (line 184) that only 11 patients were questioned about sexual violence; was this on the questionnaire or not? If so, why were only 11 patients asked about it?

In the results section, it states that 202 patients were recruited, but there is no mention early on in Results that some were excluded due to not having RC. Please include.

Figure 4 is confusing because not all categories sum to 168 (recurring vs relapsing and chronic vs non-chronic). Please clarify.

Discussion, lines 168, 194, 228- need to update the citation

Author Response

Dear Reviewer,

Thank you for your comments. Please find below the authors’ replies.

This manuscript is a survey completed about women who have recurrent cystitis at one clinic over the course of ~2 years. The goal was to characterize the patients and their management. The authors should be commended for their efforts in capturing this data so that we know more about this difficult-to-manage patient group.

My biggest concern about the manuscript is that its utility is likely limited to the authors’ country of France (and maybe just limited to their center too).

The issue addressed in our manuscript does not appear restricted to France: a survey conducted in England among patients and GPs showed that 1) management of recurrent cystitis was often unsatisfactory for women due to their lack of communication with their GP, 2) antibiotic stewardship could be improved and 3) patient empowerment was desirable for self-management of their condition (Published online 2020 Feb 11. doi: 10.3399/bjgp20X708173). Similarly, a study conducted in 5 countries highlighted the impact of RC on women’s quality of life (https://doi.org/10.1080/14737167.2017.1359543). In Canada, recent efforts have resulted in better characterization of patients and improved antibiotic stewardship with fewer and more appropriate prescriptions, although ciprofloxacin remains  the second most prescribed antibiotic for RC (doi: 10.5489/cuaj.7697). Furthermore, among patients who visited the ID department, some came from various regions of France, but also from Spain or Italy, expressing similar complaints.

My next big concern is that the Methods are not very clear. In the abstract, the authors claim this is a cohort study, but it seems like the questionnaire was only completed one time for each patient, which is more indicative of a cross-sectional design. Additionally, were the questionnaires completed by asking at least some of the questions to the patients? Or solely by the knowledge of the physicians? This should be clarified.

The methods section has been clarified. It is a cross-sectional design. The questionnaires were completed with the patient.

My third concern is the first paragraph in Results coupled with the ethical approval statement. If at least some of the information on the questionnaire needed to be completed by talking with the patient, then why were 54 responses included when they only used the medical files? And why were the questionnaires not completed with the patients? This concerns me that these patients might not have wanted their data collected and analyzed, even in anonymous form. This is especially concerning with the statement that ethical approval was not required. Was ethical approval sought and deemed “exempt” or not required? Or was it not sought at all? Assuming ethical approval was sought and deemed not required, then one way to modify this is to re-run the analysis with only the 148 questionnaires that were completed with the patients. I’m further confused by the statement in Discussion (line 184) that only 11 patients were questioned about sexual violence; was this on the questionnaire or not? If so, why were only 11 patients asked about it?

All patients spent an hour with the infectious diseases physician. The questionnaire had either been completed before the consultation by the patient or else was discussed with and completed by the physician with the patient present, during the consultation. This is because the questionnaire had not been finalised for the first 54 patients, while the remainder could complete it beforehand.

Specific ethical approval was not required because all patients presenting at the institution systematically sign a document stating their data may be used anonymously for research purposes. This has now been specified in the manuscript.

Sexual violence was not specifically investigated in the questionnaire, women were asked whether they had ever been assaulted during their lifetime. Since it is a difficult issue, the investigator was not comfortable with raising it with each patient as should have been done.

In the results section, it states that 202 patients were recruited, but there is no mention early on in Results that some were excluded due to not having RC. Please include.

Since this was a specialized consultation, all patients were referred with a diagnosis of RC, and no patients were excluded. Subsequently, 17% of patients were found not to suffer from RC: some had recurring positive urine cultures but were symptom-free, therefore considered as cases of bladder colonization, others had negative urine cultures, their symptoms resulting from menopause genitourinary syndrome. These patients were included because the aim of the study was to assess the adequacy of patient management when they had previously been mistakenly diagnosed with RC. The specialized consultation was thus an opportunity to clarify the diagnosis and to educate patients on the nature and management of their condition.

These patients are included in the flow-chart.

Figure 4 is confusing because not all categories sum to 168 (recurring vs relapsing and chronic vs non-chronic). Please clarify.

Figure 4 has been clarified: Data were missing for 23 patients regarding the distinction between recurring (with different bacterial agents) and relapsing (with the same bacterial agent) cystitis. The distinction between chronic (over 10 episodes a year) and non-chronic cystitis was not clear-cut for all patients due to recall bias.

Discussion, lines 168, 194, 228- need to update the citation

Citations have been updated: line 168: SPILF recommendations, ref #7; line 194, ref #21; line 228, ref #27.

Minor errors were corrected on Tables 1 & 2.

Missing captions were added  

Reviewer 2 Report

The data collected by the authors are very valuable and the intention of this study is very good. However I would add a few suggestions.

  1. The nature of the study must be mentioned in the methods section.
  2. The developed questionnaire should be detailed in the method section.
  3. What was the statistical method used?
  4. The timing of the study is not clear in relation to data collection. Please restructure the results in the order of their acquisition.
  5. For the accuracy of the data it would be useful to use a control group on which the questionnaire was used.
  6. Lines 85-87: Depression is not quite a subjective symptom. Was it documented by a psychiatrist? Subjective accusations should not be mixed with illnesses.
  7. Lines 97-99: how did you choose and monitor the patients? why only 65% and why only 20% were followed up? is it missing data that needs to be detailed?
  8. Lines 103-105, 111-113: By what fully effective criteria? please define.
  9. How were antimicrobial agents administered? empirically? please specify.
  10. Lines 106-108: please define appropriate.
  11. The full explanation of the abbreviated words should be put in the text, not just in the caption.
  12. Please add a caption for the Figure 1.
  13. The lines 122-124 are vague. There are many missing data that are not mentioned. Please detail the spilf recommendations for not doing those investigations. Did patients report that those investigations were not done or is there concrete information that doctors followed the guidelines?
  14. In the results section, the clinical signs, self-perceived psychological profile and hormonal status are not evaluated as written in the methods section.
  15. Line 145: how was it inappropriate? the same for the line 148.
  16. Line 151: how come the only factor was sexual intercourse for 44% of cases? for the remaining 56% with post-coital RC, the sexual intercourse was not a risk factor?
  17. Figure 4 doesn't make much sense. It needs to be redone and explained.
  18. The conclusion is not anchored. What exactly are the management failures? it is not apparent from the results. No statistical correlation between antibiotic prescription and outcomes. Tailored how? Do you have any recommendations?
  19. I did not find any of the data on quality of life impairment, psychological impact, that you mention in the conclusions. What does the ecological and autonomous approach refer to?

Author Response

Dear Reviewer,

Thank you for your comments. Please find below the authors’ replies.

Comments and Suggestions for Authors

The data collected by the authors are very valuable and the intention of this study is very good. However I would add a few suggestions.

  1. The nature of the study must be mentioned in the methods section.

This has been mentioned: tis is a cross-sectional single centre study.

    2. The developed questionnaire should be detailed in the method section.

The questionnaire has been added as an appendix or supplementary material

 3. What was the statistical method used?

Data were entered and analysed, and graphs constructed using MS Excel software. Descriptive statistical analysis was performed with variables shown as N and %.

4. The timing of the study is not clear in relation to data collection. Please restructure the results in the order of their acquisition.

Questionnaires were completed with the patient during the first consultation.  

5. For the accuracy of the data it would be useful to use a control group on which the questionnaire was used.

Given that these patients were referred to a specialized unit with a diagnosis of RC, it was not feasible in this context to include a control group. Other publications regarding patient profiles, clinical situations and management report results similar to ours, thus supporting our findings, namely in terms of anxiety, a characteristic which appears particularly frequent among women with cystitis. Ref 14

6. Lines 85-87: Depression is not quite a subjective symptom. Was it documented by a psychiatrist? Subjective accusations should not be mixed with illnesses.

Patients stated they were anxious and seven percent said they took antidepressant medication.

7. Lines 97-99: how did you choose and monitor the patients? why only 65% and why only 20% were followed up? is it missing data that needs to be detailed?

This is not missing data. Although urinalysis and culture should be performed at each episode of cystitis, systematic follow-up is not recommended for RC. Indeed, it may reveal simple colonization following treatment which can be an unnecessary cause of anxiety and renewed antimicrobial prescription.

8. Lines 103-105, 111-113: By what fully effective criteria? please define.

These are subjective clinical criteria, essentially symptom alleviation.

Lines 111-113: The inadequacy of treatment as per the French language ID society’s guidelines

9. How were antimicrobial agents administered? empirically? please specify.

Antimicrobial agents were usually prescribed empirically after urinalysis and culture, occasionally subsequently adapted to antibiotic susceptibility

10. Lines 106-108: please define appropriate.

Appropriate with regard to the French language ID society’s guidelines, namely avoiding critically important antimicrobial agents (fluoroquinolones, 3rd generation cephalosporins)

11. The full explanation of the abbreviated words should be put in the text, not just in the caption.

This has been corrected.

12. Please add a caption for the Figure 1.

The caption has been added

13. The lines 122-124 are vague. There are many missing data that are not mentioned. Please detail the spilf recommendations for not doing those investigations. Did patients report that those investigations were not done or is there concrete information that doctors followed the guidelines?

Women with recurrent cystitis who are not at risk for complications should undergo a clinical pelvic and ultrasound examination. In case of risk factors for complications, investigations should be decided in accordance with the clinical situation. Among our patients, many had unnecessarily undergone a cystoscopy, which was not recommended in their case. Furthermore, urologists had not given patients the results of the procedure. This has been added in the discussion.

14. In the results section, the clinical signs, self-perceived psychological profile and hormonal status are not evaluated as written in the methods section.

Clinical signs have been added in the results, as well as the proportion of women reporting anxiety. 

15. Line 145: how was it inappropriate? the same for the line 148.

Contrary to the SPILF guidelines recommending avoidance of critically important antibiotics, i.e. 3rd generation cephalosporins and fluoroquinolones, as well as Co-trimoxazole, these were nevertheless prescribed to a significant number of patients.

16. Line 151: how come the only factor was sexual intercourse for 44% of cases? for the remaining 56% with post-coital RC, the sexual intercourse was not a risk factor?

It was not the exclusive trigger factor.

17. Figure 4 doesn't make much sense. It needs to be redone and explained.

Information was missing so that some patients could not be allotted to a particular category. This has been specified in the caption.

18. The conclusion is not anchored. What exactly are the management failures? it is not apparent from the results. No statistical correlation between antibiotic prescription and outcomes. Tailored how? Do you have any recommendations?

The conclusion has been re-worded. Treatment of cystitis is relatively simple and straightforward. The objective is to treat RC using an approach as respectful as possible of the microbial ecosystem without favouring secondary infection, i.e. to solve the problem as best as possible with the least risk. At the same time, this approach must take into account that women currently have to wait 5 to 7 days from the time they take their urine sample to the lab, obtain the result, see their GP to get a prescription, a complex and lengthy process which could be shortened if they were given more autonomy including advice for self-care and prevention, . Furthermore, it is important to consider the highly multifactorial aspect of cystitis that should thus require a tailored and holistic approach, namely taking into account its psychological aspects.

19. I did not find any of the data on quality of life impairment, psychological impact, that you mention in the conclusions. What does the ecological and autonomous approach refer to?

Complaints expressed by women suffering from cystitis (pain, urinary urgency and frequency, interference with their sexual life) and the accompanying anxiety stated by many patients in this study illustrate in our view how the condition impairs their quality of life. As for the ecological and autonomous approach, please refer to item 18.

Minor errors were corrected on Tables 1 & 2.

Missing captions were added 

Round 2

Reviewer 1 Report

This revision has incorporated many of the suggested changes from the reviewers. However, there are still a few changes that have not yet been made.

-Despite the clarification in the Methods section, the abstract still states that the study is a cohort study, which is inaccurate.

-The response that the authors provided about lack of generalizability outside of France is well written and should be included in the manuscript so that other readers get a better sense of the worldwide applicability

-There are still issues with citations (like in line 249)

Author Response

Dear Reviewer,

Thank you for your comments.

This revision has incorporated many of the suggested changes from the reviewers. However, there are still a few changes that have not yet been made.

-Despite the clarification in the Methods section, the abstract still states that the study is a cohort study, which is inaccurate.

This has been corrected

-The response that the authors provided about lack of generalizability outside of France is well written and should be included in the manuscript so that other readers get a better sense of the worldwide applicability

This is now included in the discussion

-There are still issues with citations (like in line 249)

The citation is correct, although the title of the paper may not suggest this, the comment concerning pyelonephritis was found in that paper.

Reviewer 2 Report

 Despite the guidance given to the authors to improve the quality of the presented study, i did not observe the major changes requested. The suggestions were well pointed and clearly formulated, but did not lead to any result. This study has major flaws in the study design and presentation.

Author Response

Dear Reviewer

Despite the guidance given to the authors to improve the quality of the presented study, i did not observe the major changes requested. The suggestions were well pointed and clearly formulated, but did not lead to any result. This study has major flaws in the study design and presentation.

We apologize for the incomplete revision which was a mistake, as we did not submit the right version of the revised manuscript. We have now detailed the method and statistical analysis. With regards to your prior comments and suggestions, we consider that we have now improved the manuscript

Haut du formulaire

  1. The nature of the study must be mentioned in the methods section.

This has been mentioned: this is a cross-sectional single centre study.

  1. The developed questionnaire should be detailed in the method section.

The questionnaire has been added as an appendix or supplementary material

  1. What was the statistical method used?

Data were entered and analysed, and graphs constructed using MS Excel software. Descriptive statistical analysis was performed with variables shown as N and %.

  1. The timing of the study is not clear in relation to data collection. Please restructure the results in the order of their acquisition.

Questionnaires were completed with the patient during the first consultation.

  1. For the accuracy of the data it would be useful to use a control group on which the questionnaire was used.

Given that these patients were referred to a specialized unit with a diagnosis of RC, it was not feasible in this context to include a control group. Other publications regarding patient profiles, clinical situations and management report results similar to ours, thus supporting our findings, namely in terms of anxiety, a characteristic which appears particularly frequent among women with cystitis. Ref 14

  1. Lines 85-87: Depression is not quite a subjective symptom. Was it documented by a psychiatrist? Subjective accusations should not be mixed with illnesses.

Patients stated they were anxious and seven percent said they took antidepressant medication. This was a questionnaire completed and discussed with the patient.

  1. Lines 97-99: how did you choose and monitor the patients? why only 65% and why only 20% were followed up? is it missing data that needs to be detailed?

Patients were referred, not chosen, and offered to participate. It was a specialized infectious diseases consultation, most continued being monitored by their own general practitioner  following referral.

This is not missing data. Although urinalysis and culture should be performed at each episode of cystitis, systematic follow-up is not recommended for RC. Indeed, it may reveal simple colonization following treatment which can be an unnecessary cause of anxiety and renewed antimicrobial prescription.

  1. Lines 103-105, 111-113: By what fully effective criteria? please define.

These are subjective clinical criteria, essentially symptom alleviation.

Lines 111-113: The inadequacy of treatment as per the French language ID society’s guidelines

  1. How were antimicrobial agents administered? empirically? please specify.

Antimicrobial agents were usually prescribed empirically after urinalysis and culture, occasionally subsequently adapted to antibiotic susceptibility

  1. Lines 106-108: please define appropriate.

Appropriate with regard to the French language ID society’s guidelines, namely avoiding critically important antimicrobial agents (fluoroquinolones, 3rd generation cephalosporins)

  1. The full explanation of the abbreviated words should be put in the text, not just in the caption.

This has been corrected.

  1. Please add a caption for the Figure 1.

The caption has been added

  1. The lines 122-124 are vague. There are many missing data that are not mentioned. Please detail the spilf recommendations for not doing those investigations. Did patients report that those investigations were not done or is there concrete information that doctors followed the guidelines?

Women with recurrent cystitis who are not at risk for complications should undergo a clinical pelvic and ultrasound examination. In case of risk factors for complications, investigations should be decided in accordance with the clinical situation. Among our patients, many had unnecessarily undergone a cystoscopy, which was not recommended in their case. Furthermore, urologists had not given patients the results of the procedure. This has been added in the discussion.

  1. In the results section, the clinical signs, self-perceived psychological profile and hormonal status are not evaluated as written in the methods section.

Clinical signs have been added in the results, as well as the proportion of women reporting anxiety.

  1. Line 145: how was it inappropriate? the same for the line 148.

Contrary to the SPILF guidelines recommending avoidance of critically important antibiotics, i.e. 3rd generation cephalosporins and fluoroquinolones, as well as Co-trimoxazole, these were nevertheless prescribed to a significant number of patients.

  1. Line 151: how come the only factor was sexual intercourse for 44% of cases? for the remaining 56% with post-coital RC, the sexual intercourse was not a risk factor?

It was not the exclusive trigger factor.

  1. Figure 4 doesn't make much sense. It needs to be redone and explained.

Information was missing so that some patients could not be allotted to a particular category. This has been specified in the caption.

  1. The conclusion is not anchored. What exactly are the management failures? it is not apparent from the results. No statistical correlation between antibiotic prescription and outcomes. Tailored how? Do you have any recommendations?

The management failures are the inappropriate antibiotic prescriptions, unnecessary investigations, and lack of consideration of the condition’s impact on patients’ quality of life. The conclusion has been re-worded. Treatment of cystitis is relatively simple and straightforward. The objective is to treat RC using an approach as respectful as possible of the microbial ecosystem without favouring secondary infection, i.e. to solve the problem as best as possible with the least risk. At the same time, this approach must take into account that women currently have to wait 5 to 7 days from the time they take their urine sample to the lab, obtain the result, see their GP to get a prescription, a complex and lengthy process which could be shortened if they were given more autonomy including advice for self-care and prevention, . Furthermore, it is important to consider the highly multifactorial aspect of cystitis that should thus require a tailored and holistic approach, namely taking into account its psychological aspects.

  1. I did not find any of the data on quality of life impairment, psychological impact, that you mention in the conclusions. What does the ecological and autonomous approach refer to?

Complaints expressed by women suffering from cystitis (pain, urinary urgency and frequency, interference with their sexual life) and the accompanying anxiety stated by many patients in this study illustrate in our view how the condition impairs their quality of life. However, we agree that a QoL assessment scale would have been appropriate. As for the ecological and autonomous approach, please refer to item 18.

Minor errors were corrected on Tables 1 & 2.

Missing captions were added

Round 3

Reviewer 2 Report

The authors made the most changes as recommended. I am satisfied with the final version and i approve it to be published.